# Tigecycline Sensitivity Reduction in *Escherichia coli* Due to Widely Distributed *tet*(A) Variants

**DOI:** 10.3390/microorganisms11123000

**Published:** 2023-12-18

**Authors:** Shan Zhang, Mingquan Cui, Dejun Liu, Bo Fu, Tingxuan Shi, Yang Wang, Chengtao Sun, Congming Wu

**Affiliations:** 1National Key Laboratory of Veterinary Public Health and Safety, College of Veterinary Medicine, China Agricultural University, Beijing 100193, China; bs20203050454@cau.edu.cn (S.Z.); liudejun@cau.edu.cn (D.L.); fubo@cau.edu.cn (B.F.); s20223050919@cau.edu.cn (T.S.); wangyang@cau.edu.cn (Y.W.); 2Key Laboratory of Animal Antimicrobial Resistance Surveillance, Ministry of Agriculture and Rural Affairs, College of Veterinary Medicine, China Agricultural University, Beijing 100193, China; 3China Institute of Veterinary Drug Control, Beijing 100081, China; qinshihuang320@163.com

**Keywords:** *tet*(A) variants, tigecycline resistance, *Escherichia coli*

## Abstract

Despite scattered studies that have reported mutations in the *tet*(A) gene potentially linked to tigecycline resistance in clinical pathogens, the detailed function and epidemiology of these *tet*(A) variants remains limited. In this study, we analyzed 64 *Escherichia coli* isolates derived from MacConkey plates supplemented with tigecycline (2 μg/mL) and identified five distinct *tet*(A) variants that account for reduced sensitivity to tigecycline. In contrast to varied tigecycline MICs (0.25 to 16 μg/mL) of the 64 *tet*(A)-variant-positive *E. coli* isolates, gene function analysis confirmed that the five *tet*(A) variants exhibited a similar capacity to reduce tigecycline sensitivity in DH5α carrying pUC19. Among the observed seven non-synonymous mutations, the V55M mutation was unequivocally validated for its positive role in conferring tigecycline resistance. Interestingly, the variability in tigecycline MICs among the *E. coli* strains did not correlate with *tet*(A) gene expression. Instead, a statistically significant reduction in intracellular tigecycline concentrations was noted in strains displaying higher MICs. Genomic analysis of 30 representative *E. coli* isolates revealed that *tet*(A) variants predominantly resided on plasmids (*n* = 14) and circular intermediates (*n* = 13). Within China, analysis of a well-characterized *E. coli* collection isolated from pigs and chickens in 2018 revealed the presence of eight *tet*(A) variants in 103 (4.2%, 95% CI: 3.4–5.0%) isolates across 13 out of 17 tested Chinese provinces or municipalities. Globally, BLASTN analysis identified 21 *tet*(A) variants in approximately 20.19% (49,423/244,764) of *E. coli* genomes in the Pathogen Detection database. These mutant *tet*(A) genes have been widely disseminated among *E. coli* isolates from humans, food animals, and the environment sectors, exhibiting a growing trend in *tet*(A) variants over five decades. Our findings underscore the urgency of addressing tigecycline resistance and the underestimated role of *tet*(A) mutations in this context.

## 1. Introduction

Antimicrobial drugs have significantly advanced global public health, animal health, and food safety, with tetracyclines emerging as one of the key contributors. Due to their broad-spectrum activity, high oral bioavailability, and cost-effectiveness, tetracyclines have been long and widely used in both human and animal healthcare worldwide [1,2]. Consequently, resistance to tetracyclines in *Escherichia coli* has become widespread, with reports indicating that as high as 60% of global *E. coli* isolates from animals were resistant to tetracycline, surpassing the resistance rates of isolates from environmental (41%) and clinical (37.3%) sources [3]. In China, the resistance rate of *E. coli* from animal sources to tetracycline has consistently exceeded 80% over multiple years [4]. Currently, there are 62 different genes found in various genera of bacteria conferring tetracycline resistance, including 36 tetracycline efflux genes, 13 ribosome protective protein-encoding genes, and 13 enzymatic inactivation genes [5]. Among these, the tetracycline efflux gene, *tet*(A), emerged after the clinical usage of tetracycline [6] and is currently prevalent in bacterial populations found in humans and animals alike [7], representing a profound impact on the clinical efficacy of tetracyclines.

In response to this issue, tigecycline, the first member of the glycylcycline class of tetracyclines, was designed and first approved for use in human medicine in the US in 2005, particularly in cases of life-threatening infections [8]. The N, N-dimethylglycylamido moiety at position 9 of the tetracycline nucleus possesses potent antimicrobial activity against tetracycline-resistant strains expressing either ribosomal protection or efflux determinants [9]. As such, tigecycline is unable to be recognized as a substrate of the efflux pump Tet(A) [10]. However, despite its initial efficacy, resistance to tigecycline has recently emerged. Surveillance studies have reported relatively high levels of tigecycline resistance in numerous Enterobacteriaceae isolates, primarily mediated by the oxidoreductase coding gene *tet*(X) [11,12,13] and the non-specific active Resistance-Nodulation-Division efflux pump coding gene *tmexCD-toprJ* [14]. Notwithstanding that these plasmid-mediated genes are prevalent in food animals, their presence in human populations appears to be rare.

In contrast to the rarity of those high-level tigecycline resistance genes, the tetracycline efflux gene *tet*(A) is widely distributed among bacteria found in both human clinical settings [15] and food animals [16]. Clinical cases of infections caused by *Klebsiella pneumoniae* [17] and *Salmonella enterica* [18] have been reported with low-level tigecycline resistance, wherein the bacteria harbored a mutant *tet*(A). This mutant *tet*(A) has presumed the ability to recognize and efflux tigecycline, contributing to clinical treatment challenges. Nevertheless, the detailed function and clinical implications of these *tet*(A) mutations in mediating tigecycline resistance remain neglected and largely unknown. Given the clinical importance of tigecycline, the wide prevalence of *tet*(A), and the emerging risk of *tet*(A) mutants conferring treatment challenges, comprehensive surveillance is strongly recommended to elucidate their function and epidemiology in isolates from humans and animals.

To elucidate the function of *tet*(A) variants in mediating tigecycline resistance and their presence in animals and humans, we analyzed a collection of *E. coli* strains from food animals across China and the *E. coli* genomes in the Pathogen Detection database. We ascertained that a variety of *tet*(A) variants could mediate reduced tigecycline sensitivity in DH5α carrying pUC19 clones. These mutant *tet*(A) genes are predominantly plasmid-borne and widely disseminated among *E. coli* isolates from food animals and human clinics worldwide.

## 2. Materials and Methods

### 2.1. Bacterial Strains

During surveillance of the tigecycline resistance gene *tet*(X) in a national collection of *E. coli* strains from food animals across China [12,19], we obtained 64 *E. coli* isolates from MacConkey plates supplemented with tigecycline (2 µg/mL). These isolates are unduplicated and originate from pigs (*n* = 20) and chickens (*n* = 44) from 28 provinces in China. PCR analysis indicated the 64 *E. coli* isolates are negative for *tet*(X) and *tmexCD-toprJ*, which are the major reported transferable tigecycline resistance determinants. To identify any potential mechanisms in mediating tigecycline resistance or reduced sensitivity, the 64 *E. coli* isolates were included in the current study and analyzed by the following experiments. To elucidate the presence of *tet*(A) variants in *E. coli* isolates from food animals, we analyzed a well-characterized *E. coli* collection (*n* = 2475) isolated from pigs and chickens in 2018 and identified 103 *E. coli* isolates harboring *tet*(A) variant, 87 of which were co-harboring the gene *tet*(X).

### 2.2. Antimicrobial Susceptibility Testing

Minimum inhibitory concentrations (MIC) were determined for the 64 *E. coli* isolates using broth microdilution in fresh Mueller-Hinton broth (Cation-adjusted, BD Ltd., Franklin Lakes, NJ, USA) following the guidelines of the Clinical and Laboratory Standards Institute documents (CLSI) with the incubation conditions of 35 ± 2 °C, ambient air, and 16–20 h [20]. The tested antibiotics included tetracycline, minocycline, tigecycline, and omadacycline. Tetracycline and minocycline susceptibility were interpreted as susceptible (MIC ≤ 4 μg/mL), intermediate (MIC = 8 μg/mL), and resistant (MIC ≥ 16 μg/mL) according to the CLSI. Susceptibility of tigecycline was interpreted according to US Food and Drug Administration (FDA) breakpoints (susceptible: MIC ≤ 2 μg/mL, intermediate: MIC = 4 μg/mL, and resistant: MIC ≥ 8 μg/mL). And omadacycline’s breakpoint, according to the FDA, is the same as that of tetracycline above. Reference strain *E. coli* ATCC 25922 served as the quality control.

### 2.3. Conjugation Assay

To test the transferability of the tigecycline-resistant determinant, conjugative assays were performed. The sodium-azide-resistant *E. coli* J53 was used as the recipient, while a sodium-azide-susceptible but tigecycline-resistant *E. coli* strain LNp148 (tigecycline MIC = 8 μg/mL) served as the donor strain. The LNp148 and J53 strains were mixed in equal proportions and cultured at 37 °C for 16 h. Transconjugants were selected by plating the mixture on LB agar supplemented with tigecycline (2 μg/mL) and sodium azide (100 μg/mL).

### 2.4. Functional Metagenomics

The plasmid DNA extracted from transconjugant J53, which contained the plasmid from LNp148, was sheared into fragments of approximately 3 kb using a Convaris 220 sonicator. These DNA fragments were then cloned into a colony vector pZE21 MCS-1 in *E. coli* DH5α. Transformants potentially carrying the tigecycline-resistant determinant were selected on LB agar plates containing 1 μg/mL of tigecycline and 50 μg/mL of kanamycin. Subsequently, the resulting fragments were PCR-amplified and subjected to sequencing.

### 2.5. Cloning of tet(A) Variants

Mutations within the *tet*(A) gene were characterized by comparing the sequences with the corresponding regions in plasmid RP1 (GenBank accession number X00006) using the online NCBI BLASTN tool. To investigate the phenotype variations mediated by these mutations, we cloned all *tet*(A) variants into plasmid pUC19 (Qing Ke, Nanjing, China) using primers listed in Appendix A. The *tet*(A) variants were expressed under the control of a *lac* promoter. These plasmids were then transformed into *E. coli* DH5α. Point mutations of synthetic *tet*(A) strain DH5α-pUC19-Tet(A) were made by recombination using a Seamless Cloning Kit (primers for cloning are listed in Appendix A).

### 2.6. Whole-Genome Sequencing and Bioinformatics Analysis

All 64 *E. coli* isolates were subjected to whole-genome sequencing analysis. In brief, the total DNA was extracted, and the sequence libraries were constructed using the KAPA Hyper Prep Kit (Kapa Biosystems, Boston, MA, USA). These libraries were sequenced on the Illumina HiSeq X 10 platform (Annoroad, Beijing, China). Draft assemblies of the cleaned reads were generated using SPAdes version 3.9.0 [21]. To determine the genetic location of the *tet*(A) variants (whether on plasmids or chromosomes), 30 representative isolates, including all the five *tet*(A) variants, were selected and sequenced on the MinION platform (Oxford Nanopore Technologies, Oxford, UK). Genome assemblies combining the Illumina and MinION sequences were generated using Unicycler v.0.4.8-beta [22]. Contigs containing the *tet*(A) gene were identified and extracted from assemblies using contig-puller (http://github.com/kwongj/contig-puller) (accessed on 25 July 2021), followed by cyclization checks. Circular contigs containing plasmid replicons (>95% identity and >90% coverage according to PlasmidFinder [23]) were considered plasmid-borne, while circular contigs with no replicons were considered circular intermediates.

We downloaded all *E. coli* assemblies from the NCBI Pathogen Detection database (https://www.ncbi.nlm.nih.gov/pathogens/isolates/) (accessed on 31 May 2023, *n* = 244,764) and manually curated each isolate for its origin based on the records of “Host”, “Isolation source”, and “Isolation type”. *E. coli* isolates were classified as having “human” or “animal” origins based on any clear indication from the records; otherwise, they were classified as “environmental/other” isolates. The DNA sequences downloaded were used to perform BLAST for the identification of the type of *tet*(A) variant. All identified *tet*(A) variants protein sequences were aligned with CLUSTALW software (https://www.genome.jp/tools-bin/clustalw/) (accessed on 28 July 2023), and cluster heatmap of Tet(A) variants protein was depicted by ChiPlot (https://www.chiplot.online/) (accessed on 28 July 2023). The structure-based sequences alignments and 2D structure were performed using ESPript (http://espript.ibcp.fr/ESPript/cgi-bin/ESPript.cgi) (accessed on 28 July 2023).

### 2.7. RNA Extraction and RT-PCR Analysis

From the 30 representative *E. coli* isolates sequenced on the MinION platform, all 26 *E. coli* isolates harboring *tet*(A)v1 variants were selected and analyzed for phenotype variations. However, 3 of the 26 *E. coli* isolates were excluded due to RNA contamination. *E. coli* isolates containing *tet*(A)v1 variants were incubated in LB for 5 h. Subsequently, 2 mL of such cultures were supplemented with 0.5 µg/mL tigecycline and further incubated for an additional 1 h. Concurrently, another 2 mL culture was incubated without tigecycline for 1 h. Total RNA from these two cultures was extracted using the EASYsoin Plus kit (Aidlab). For reverse transcription, 1 µg of the extracted RNA was used, and the process was carried out using the Evo M-MLV RT premix for qPCR (gDNA wiper) Kits (Accurate Biology). The relative expression of *tet*(A), *tet*(R), and other genes was normalized using 16S rRNA with three technical replicates of each isolate. Primer sequences used have been presented in Appendix A.

### 2.8. Accumulation of Tigecycline within E. coli Harboring tet(A) Variant

To investigate whether the varying tigecycline MIC in *tet*(A) variant *E. coli* is attributed to differences in intracellular drug concentration, we assessed tigecycline accumulation following previous reports [14,24]. *E. coli* isolates containing the *tet*(A) variant were cultured in LB medium with shaking (200 rpm) at 37 °C to late logarithmic phase (for 8 h). Subsequently, bacteria cells were harvested by centrifugation at 4500× *g* for 10 min at 4 °C, washed twice with phosphate-buffered saline (PBS, pH 7.0), and resuspended in PBS to an optical density (OD600) of 1.0. From this suspension, nine aliquots of 300 µL each were obtained. In the first three 300 µL aliquots, tigecycline was added at a final concentration of 10 µg/mL. In the next three 300 µL aliquots, tigecycline (10 µg/mL) and Carbonyl Cyanide 3-ChloroPhenylhydrazone (CCCP = 10 µg/mL) were added. The remaining three 300 µL aliquots were used as blank controls without any drug supplementation. After 15 min of incubation at 37 °C, 300 µL of ice-cold PBS was immediately added to each aliquot and centrifuged at 6000 rpm for 10 min at 4 °C. Bacterial cell pellets were washed once with 1 mL of ice-cold PBS. To lyse the samples, 300 μL of water was added and mixed with each pellet, followed by three freeze–thaw cycles using liquid nitrogen and a water bath at 50 °C. The lysates were then centrifuged at 12,000× *g* for 3 min to collect the supernatants. Additionally, to ensure complete lysis of the bacteria, the residual debris was resuspended in 200 μL of acetonitrile and centrifuged to collect the supernatants. All supernatants were filtered using a 0.22 μm filter membrane. Finally, all supernatants were analyzed using the AB SCIEX Triple Quad^TM^ 7500 mass spectrometer (ABSciex, Foster City, CA, USA). The mobile phase consisted of 0.1% formic acid in water (A) and 0.1% formic acid-methanol:acetonitrile = 2:8 (B). Separation occurred on a C18 column at a flow rate of 0.4 mL/min. The linear gradient was as follows: 0–1 min, 95% A; 1.5–2.5 min, 50% A; 3.0–4.0 min, 5% A; and 4.1–5.0 min, 95% A. Tigecycline accumulation was quantitatively detected using multiple reaction monitoring (MRM) with positive electrospray ionization and the *m*/*z* 586.1→513.1 transition.

### 2.9. Statistical and Correlation Analyses

All experimental data were statistically analyzed using GraphPad Prism 9.0 software. The gene expression level and the concentration of intracellular tigecycline of different MIC groups were presented as mean ± SD. The Kruskal–Wallis test was adopted as the post-hoc test to calculate *p*-values. Differences with *p* < 0.05 were considered significant. Significance levels were indicated by asterisks: * *p* < 0.05, ** *p* < 0.01, *** *p* < 0.001.

## 3. Results

### 3.1. Mutations of the tet(A) Gene Decrease Tigecycline Sensitivity in E. coli

Antimicrobial susceptibility testing confirmed that the 64 *E. coli* isolates exhibited varied MICs against tigecycline, ranging from 0.25 to 16 μg/mL, with 15 of the 64 *E. coli* isolates found to be resistant to tigecycline (Figure 1A). Conjugation experiment suggests potential tigecycline resistance determinants could be transferred from a wild *E. coli* strain (LNp148, tigecycline MIC = 8 μg/mL) into the recipient strain (*E. coli* J53, tigecycline MIC = 0.25 μg/mL), giving the transconjugant a 32-fold increase in tigecycline MIC (8 μg/mL). Based on the plasmid DNA of the transconjugant, functional metagenomics singled out the tetracycline efflux gene *tet*(A), potentially attributing to the tigecycline resistance of the wild *E. coli* strain (LNp148). BLAST analysis of the *tet*(A) gene from LNp148 genome assembly identified mutations compared to the original *tet*(A) gene from plasmid RP1 (GenBank accession number X00006). Moreover, *tet*(A) genes were detected in all 64 *E. coli* isolates.

Among the 64 *tet*(A)-positive *E. coli* isolates, *tet*(A)v1 was the most predominant, present in 60 of the 64 isolates. The remaining four isolates harbored four additional variants designated *tet*(A)v2 to *tet*(A)v5. Additionally, *tet*(A)-1, the previously reported allele of *tet*(A), was also found in 6 of the 60 isolates harboring *tet*(A)v1 (Appendix A). The five variants exhibited a shared non-synonymous mutation profile of I5R, V55M, I75V, T84A, S201A, F202S, and V203F, and also a specific mutation of G126S for Tet(A)v2, A93T, and S251A for Tet(A)v3, R326M for Tet(A)v4. Tet(A)v5 was the variant of Tet(A)-1 with a mutation of M25I.To confirm whether mutations in *tet*(A) could confer resistance to tigecycline, we cloned the five *tet*(A) variants into pUC19 plasmids and transformed them into *E. coli* DH5α. The tigecycline MICs of the transformants were 8- to 16-fold higher than that of *E. coli* DH5α carrying pUC19 alone (Figure 1B). This increase was higher than the 4-fold change mediated by the wild-type *tet*(A) gene (from 0.25 to 1 μg/mL). Generally, the five Tet(A) proteins displayed a mutation profile consisting of I5R, V55M, I75V, T84A, S201A, F202S, and V203F. To investigate the function of the seven shared mutations on tetracyclines, five mutants were generated based on point mutation of the synthetic *tet*(A) sequence. Compared to DH5α-pUC19-Tet(A), DH5α-pUC19-Tet(A)-V55M exhibited a tigecycline MIC increase from 1 µg/mL to 2 µg/mL. In contrast, other mutants exhibited a relatively little effect on tigecycline resistance. Additionally, compared to DH5α-pUC19-Tet(A), DH5α-pUC19-Tet(A)-V55M also exhibited elevated MICs of other tetracyclines, including tetracycline, minocycline, and omadacycline (Table 1).

### 3.2. Phenotype Variations Are Not Correlated with the Expression of tet(A) and tet(R)

Given the large variations in tigecycline MICs observed among the 64 *tet*(A)-variant-harboring *E. coli* (Figure 1A, Appendix A) and the generally homogeneous ability of *tet*(A) variants in the transformants (Figure 1B), we were curious about whether the phenotype variations are correlated with the expression of *tet*(A) and its regulator gene *tet*(R) in the wild strains. Based on tigecycline MICs, a total of 23 *tet*(A)v1-harboring *E. coli* strains were divided into three groups (MIC ≤ 2 µg/mL, MIC = 4 µg/mL, MIC ≥ 8 µg/mL). We first ruled out any correlation between the phenotype variations and the expression of genes known to confer tigecycline resistance, including *acrA*, *acrB*, *tolC*, *ompF,* and *ompC* (Appendix A). Moreover, we found no statistically significant difference in the relative expression of *tet*(A) and *tet*(R) between these groups, with or without 0.5 µg/mL tigecycline induction (Appendix A). Given the intracellular concentration of tigecycline in bacteria is a collective response to its efflux and influx capabilities, we thus measured the intracellular accumulation of tigecycline by HPLC-mass spectrometry with or without the induction of the efflux pump inhibitor CCCP. When the bacteria were induced by 10 µg/mL CCCP, we observed no statistically significant difference in tigecycline accumulation 15 min after tigecycline (10 µg/mL) was added (Figure 2A), indicating a generally similar influx capability of tigecycline. However, without CCCP, a statistically lower intracellular tigecycline concentration was observed in isolates of the higher MIC groups (Figure 2B). This suggests advanced efflux pumping capabilities in isolates of the higher MIC groups.

### 3.3. tet(A) Variants Are Mostly Plasmid-Borne

To figure out the genetic locations of the *tet*(A) variants, we assembled 30 isolates with different tigecycline MIC of the 64 *tet*(A)-variant harboring *E. coli* using MinION long reads and Illumina short reads sequencing. Analysis of the hybrid de novo assembly indicated the *tet*(A) variants were mostly on plasmids (*n* = 14) and circular intermediates (*n* = 13). Replicon analysis indicates the plasmids belonging to a variety of incompatibility groups, including IncFIB (*n* = 4; 108,660 bp, 121,632 bp, 121,632 bp, 74,565 bp), IncX1 (*n* = 1; 52,857 bp), and IncI1 (*n* = 1; 118,603 bp), and a handful of hybrid plasmids with multi-replicons, including IncHI2-IncHI2A (*n* = 1; 235,974 bp), IncX1-IncFII (*n* = 1; 110,652 bp), IncHI2-IncHI2A-p0111 (*n* = 1; 408,758 bp), IncFIA-IncFIB-IncFIC (*n* = 1; 178,433 bp), IncY-IncII (*n* = 1; 200,527 bp), IncFIB-IncC-p0111 (*n* = 1; 254,323 bp), IncHI2-IncHI2A-IncQ1 (*n* = 1; 236,179 bp), and IncHI2-IncHI2A-IncFIB (*n* = 1; 385,151 bp). The detected circular intermediates showed specialized sizes of 5488 bp (*n* = 12) and 6831 bp (*n* = 1). Moreover, the *tet*(A) variant was also observed on the chromosome in one *E. coli* isolate (LNp197). We further observed a phenomenon of the multi-copy status of *tet*(A) variants in certain (*n* = 3) isolates.

### 3.4. tet(A) Variants Are Widely Distributed in E. coli from Human Clinical and Animals

We then determined the distribution of *tet*(A)-variant-harboring *E. coli* among the human clinical settings and the animal sectors. Based on the previous *E. coli* collection (*n* = 2475) isolated from pigs and chickens in China in 2018 [19], we identified eight *tet*(A) variants (including the previous five) in 103 *E. coli* isolates (4.2%, 95% confidence interval: 3.4–5.0%) originating from 13 of the 17 test provinces or municipalities.

From a global perspective, BLASTN analysis identified 49,423 *tet*(A)-variant-harboring *E. coli* among 244,764 *E. coli* genomes (20.19%) in the Pathogen Detection database (as of 31, May, 2023). The *tet*(A)-variant-harboring *E. coli* was widely distributed in 120 countries over the world, including the United States (*n* = 15,680, 31.72%), China (*n* = 6129, 12.40%), the United Kingdom (*n* = 4041, 8.17%), and others. These variants shared 93.0–99.7% amino acid similarity (Figure 3A). A total of 21 *tet*(A) variants were detected, and the alignment of these *tet*(A) amino acid sequences showed a highly conserved C-terminus and central portion but a variable N-terminus (Appendix A). Over the years, we observed a growing percentage of *E. coli* genomes harboring *tet*(A)-variants (Figure 3B, Appendix A). Regardless of the multi-copy state of different *tet*(A)-variants within one strain, *tet*(A)v1 accounted for the majority and was found in 33,263 of the 49,423 *tet*(A)-variant-harboring *E. coli* genomes (Appendix A). Moreover, *tet*(A)-variant-harboring *E. coli* originated from humans (*n* = 18,975, 38.39%), animals (*n* = 17,662, 35.74%), and the environment/other sector (*n* = 12,786, 25.87%).

## 4. Discussion

While scattered studies have reported mutations in the *tet*(A) gene that could confer tigecycline resistance [17,18], the clinical impact of these *tet*(A) variants in compromising tigecycline treatment efficacy has remained largely unknown. It is evident from our data that *tet*(A) variants decrease tigecycline sensitivity in *E. coli* and are widely distributed in *E. coli* isolates from both humans and animals worldwide. Considering the recent report of an amplification of a *tet*(A) variant that could confer high-level tigecycline resistance in clinical pathogens [25], the clinical importance of *tet*(A) variants should be properly assessed. However, it is essential to note that the BLASTN analysis did not identify any *E. coli* strains harboring the original *tet*(A) gene (GenBank accession number X00006). In fact, this indicates that the current *tet*(A)v1 variant represents the predominant *tet*(A) gene among different *tet*(A) variants.

Notably, we have discerned that the variability in tigecycline resistance observed across these strains does not exhibit a strong correlation with the expression levels of the *tet*(A) gene and its regulatory counterpart, *tet*(R). Instead, our analyses have pointed towards a statistical reduction in intracellular tigecycline concentrations within isolates belonging to higher MIC groups. This suggests that other factors, potentially related to efflux or influx pumping capabilities, may exert a substantial influence on the resistance phenotype. However, we failed to find any clues from the intrinsic RND pump, AcrAB-TolC, and the outer membrane proteins. This may be because the decreased intracellular tigecycline concentration is an additive effect of different determinants, or there exists some novel affecting factors. Particularly, it is noteworthy that the results obtained from *E. coli* DH5α transformants harboring *tet*(A) variants exhibited a tigecycline MIC of 4 μg/mL. In contrast, wild-type *E. coli* strains carrying *tet*(A) variants demonstrated significantly elevated tigecycline MIC values, reaching as high as 16 μg/mL. These findings underscore the multifaceted nature of tigecycline resistance mechanisms and call for further exploration of the intricate interplay of various resistance determinants.

In agreement with a previous report on clinical *S*. *enterica* [18], the wild-type *tet*(A) gene was observed in the current study to modestly increase the tigecycline MIC in DH5α-pUC19 from 0.25 to 1 µg/mL. Furthermore, DH5α-pUC19-Tet(A)-V55M strain exhibited a tigecycline MIC of 2 µg/mL, suggesting a positive role for this mutation in conferring tigecycline resistance. The amino acid substitution V55M was modeled within helices 2, which is a component of the tetracycline transport channel. Consequently, V55M may affect the transport activity of tigecycline [26,27]. However, contrary to previous reports that a double frameshift mutation (S201A, F202S and V203F) in *Salmonella* species can decrease sensitivity to tigecycline by affecting the substrate specificity of tigecycline [18,28,29], the *tet*(A) with a double frameshift mutation in our *E. coli* DH5α (DH5α-pUC19-Tet(A)-ASF) and the previous *E. coli* isogenic mutants [30] exhibited little effect on tigecycline insensitivity. In this context, we wonder if resistance to tigecycline mediated by *tet*(A) variants may vary according to bacterial species.

It is interesting to note that 15 out of the 64 isolates displayed tigecycline MIC values exceeding 8 μg/mL, surpassing the established tigecycline resistance breakpoint. This observation is of importance, as it aligns with a growing trend of *E. coli* strains harboring *tet*(A) variants over the years. This is in accordance with the upward trend of antimicrobial resistance as previously reported, and may be partially attributed to the widespread use of antimicrobials [31]. In this context, the plasmid-borne nature of the *tet*(A) gene emerges as likely one of the contributing factors to this phenomenon, warranting further investigation.

## 5. Conclusions

In summary, this study finds that multiple *tet*(A) variants can reduce the susceptibility of *E. coli* to tigecycline, which may be related to the non-synonymous mutation V55M. These mutant *tet*(A) genes have been widely disseminated among global *E. coli* isolates from humans, food animals, and the environment sectors, exhibiting a growing trend of presence over five decades. Our findings underscore the urgency of addressing tigecycline resistance and the underestimated role of tet(A) mutations in this context.

## Figures and Tables

**Figure 1 microorganisms-11-03000-f001:**
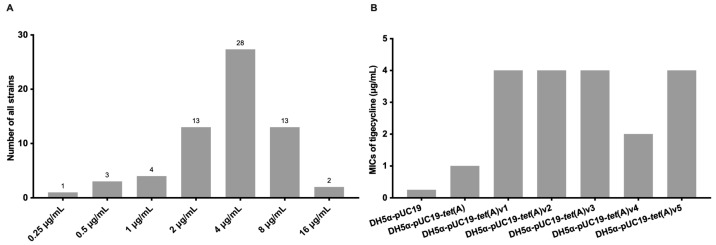
The number of tigecycline MICs observed among the 64 *tet*(A)-variant-harboring *E. coli* (**A**); Tigecycline MIC of *tet*(A) variants in the transformants (**B**).

**Figure 2 microorganisms-11-03000-f002:**
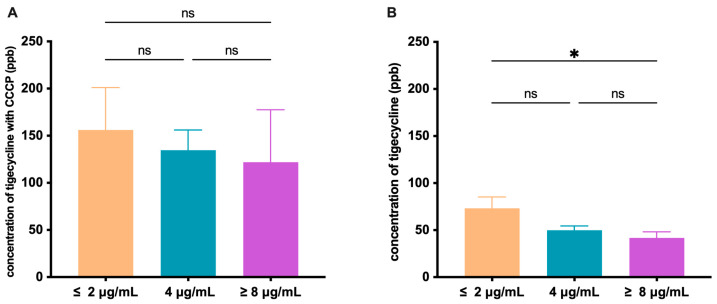
Comparison of the mean value of tigecycline accumulation in the presence (**A**) and absence (**B**) of CCCP in 23 *tet*(A)v1-positive *E. coli* at 15 min of exposure. The error bars represent the standard error of the mean. * *p* < 0.05; ns means not significant.

**Figure 3 microorganisms-11-03000-f003:**
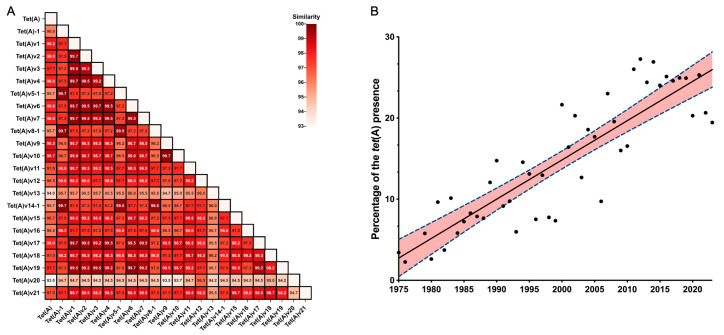
The similarity of 21 identified *tet*(A) variant proteins (**A**); Growing tendency of *tet*(A)-variant-harboring *E. coli* over time (**B**). The dots are the presence of *tet*(A) variant in different years, the solid line is a fitted linear regression line, and the red region within the dashed lines indicates 95% confidence interval for the regression line in (**B**).

**Table 1 microorganisms-11-03000-t001:** Tetracyclines’ MICs against *E. coli* with or without single mutant *tet*(A) gene.

ID	MIC ^1^ (µg/mL) of			
	TGC	TET	DOX	OMC
ATCC 25922	0.25	2	2	1
DH5α-pUC19-Tet(A)	1	20	80	4
DH5α-pUC19-Tet(A)-I5R	1	20	50	4
DH5α-pUC19-Tet(A)-V55M	2	40	120	8
DH5α-pUC19-Tet(A)-I75V	1	30	110	7
DH5α-pUC19-Tet(A)-T84A	1	30	100	5
DH5α-pUC19-Tet(A)-ASF ^2^	1	50	120	8

^1^ To differentiate the MIC of each variant against tetracycline, this study employed the concentration gradient presented in the table. TGC, tigecycline; TET, tetracycline; DOX, doxycycline; OMC, omadacycline. ^2^ DH5α-pUC19-Tet(A)-ASF carrying double frameshift mutation (S201A, F202S and V203F).

## Data Availability

The data presented in this study are available on request from the corresponding author.

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
