# Peer review of "Tigecycline Sensitivity Reduction in Escherichia coli Due to Widely Distributed tet(A) Variants"

_microorganisms, 2023, doi:10.3390/microorganisms11123000_

Round 1

Reviewer 1 Report

Comments and Suggestions for Authors

The presented report gave an excellent point of view on the urgency of addressing tigecycline resistance and the underestimated role of tet(A) mutations in this context. I recommend this manuscript for publication but after minor revisions of the abstract and especially the conclusion part. Please write and explain in the conclusion part all your findings. At the moment, when you read the abstract and conclusion it is hard to understand the main tasks, results, and their importance.

Reviewer 2 Report

Comments and Suggestions for Authors

The maniscript entitled “Tigecycline Sensitivity Reduction in Escherichia coli due to Widely Distributed tet(A) Variants” it is well structured and well written as well as conducted with considerable scientific rigor.

However, as regards the introductory part, I believe the part concerning the phenomena of resistance to tigecycline and in general to tetracyclines among the different strains of E. coli should be expanded, making use of the abundant scientific literature available on the subject.

Some data reported in the supplementary material would be worth considering if included within the main manuscript such as Figure S1 and Table S4.

Furthermore, the definition of figure 3 of the main manuscript should be improved as it is absolutely not readable.

As regards the results and the discussion, I believe that the Authors have been clear and concise and there is nothing to add or modify.

Reviewer 3 Report

Comments and Suggestions for Authors

This is an interesting manuscript about distribution of tetA variants but the authors shift the focus towards putative explanations for the different MIC values of the tested isolates irrespective of the harboured tetA variants, so the text is confusing and must highly improve.

Results must be presented in a more coherent arrangement: first isolates and mutations description and then the experiments (transformation, gene expression, etc.)

 Introduction

Line 46. Probably tigecycline has different approval dates according to country, so, please add the first one indicating where.

Lines 69-73. At the end of the introduction section, readers expect to found the objective / hypothesis tested and not a summary of the results. Please, inform the readers about the objective / hypothesis that was tested.

 Methods

Line 77. Please, specify the meaning of “chickens”. Only Gallus gallus broilers? Maybe broilers and laying hens? Others?

Lines 85-87. Since CLSI documents have not a free access to readers, please add incubation conditions (temperature, atmosphere and time) and breakpoint used.

Line 121. Please, specify the meaning of “selected strains”.

Line 182. Delete “number of”, since it is not needed.

 Results

Lines 186-189. This paragraph is not a result. Please, delete.

Line 190. “Unique” is?

Line 190. Provide the distribution of the 64 isolates among pigs and chickens.

Lines 190-201. The results regarding conjugation experiments must be located after the description of tetA variants (lines 202-218)

Line 194. Does belong this isolate to the 64 tested?

Lines 202-205. Please, check the figures: 60 isolates tetAv1 + 6 isolates tetA-1 = 66 isolates. In addition, if the other four tetA variants where detected at least in one isolate, you have to have to add 4 + 66 = 70, but only 64 isolates were studied.

So, the whole distribution of tetA variants among the 64 isolates must be added. How many isolates had the five tetA varians detected? Add data stratification by animal species.

Line 205. Before testing mutations, an in deep description of the five mutations (for instance, number of amino acid differences and location) must be added to the text summarizing the data of the supplementary figure S2. Indeed, discussion (lines 312-322) mention different mutations named V55M, S201A, F202S or V203F that the reader do not know if were detected or not by authors.

Lines 207-209. These lines contain critical information regarding the putative role of tetA variants over tigecycline MIC increase since these values are over the tigecycline breakpoint. For comparison it would interesting to add the CMI values of the isolates harbouring each of these variants.

Line 208. “fold” or “two fold” change?

Lines 217-218. Add to the M&M section the information of testing and breakpoints used for tetracycline, minocycline and omadacycline.

Lines 223-228. Had you tested the MICs according the tetA variant?

Lines 229-232. Where are the data regarding these results about gene expression of all these genes? Suplementary figure S1?

Lines 231. Having in mind the low number of isolates it is expected to found no statistically differences. Indeed, visual analysis of histogram “A” of supplementary figure S1 strongly suggest clear differences between the first column in comparison to the other two.

Lines 248-249. It is surprising for me that MIC values and not tetA variants was the criterion for selecting isolates for long-reading sequencing. Could you explain, please?

Line 251. “Mostly” is not accurate. Please, add the figures.

Lines252-258. Add the data regarding replicon distribution of tetA variants.

Lines 259-269. Specify which one was detected.

Lines 261-266. It is not clear how this information was obtained. Please, add the required data to M&M section. Where is the description of the new three tetA variants?

Lines 267-280. This paragraphs seems more discussion than results.

Line 273. “Showed”. Showing?

Line 276. The concept of “multi-copy state” is interesting. What about your studied isolates? I have not found any information on this section.

Lines 281-284. Figure 3. These figures are almost impossible to see under the current composition. I suggest that they should be presented as four single images under the supplementary data.

Discussion

Lines 286-288. Please provide some references. More important, the argument about clinical impact is not relevant since it cannot be assessed analysing the presented data.

Line 291. Please explain “amplification”.

Lines 314-318. The V55M mutation was detected in your study?.

Line 326. Specify the breakpoint.

Lines 328-394. I really do not understand this sentence.

Suplementary figure S1:

The size is so small that the figures are very difficult to see. Please, separate and add the legend, the n value of each group and the lower value for the dispersion measure.

Round 2

Reviewer 3 Report

Comments and Suggestions for Authors

I appreciate very much the effort of the authors for improving the manuscript. Nevertheless, there are still some critical points, raised in my previous comments, that can be properly explained.

  1) Preliminary comment: “Lines 69-73. At the end of the introduction section, readers expect to found the objective / hypothesis tested and not a summary of the results. Please, inform the readers about the objective / hypothesis that was tested.

Response 2: Yes, the objective of the current study was described in the end of the introduction section in lines 70-75”.

I am sorry but the new paragraph (“Nevertheless, the detailed function and clinical implications of these tet(A) mutations in mediating tigecycline resistance remains neglected and largely unknown. Given the clinical importance of tigecycline, the wide prevalence of tet(A), and the emerging risk of tet(A) mutants conferring treatment challenges, comprehensive surveillance is strongly recommended to elucidate their function and epidemiology in isolates from human and animals”) do not describe the objective. Please, add a simple but clear explanation of the goal of this study.

 2) Preliminary comment: “Lines 190-201. The results regarding conjugation experiments must be located after the description of tetA variants (lines 202-218)

Response 10: We believe there may be some misunderstandings that need clarification here. When we received the 64 E. coli isolates, we had no idea which gene accounted for the strains’ tigecycline resistance. The conjugation experiments confirmed that this potential resistance determinant is located on conjugational plasmids. Subsequently, functional metagenomics, based on the plasmid DNA from transconjugants, helped us identify that the underlying resistance determinant is tet(A) variants. In this circumstance, we think the conjugation experiment should be described before discussing the tet(A) variants.”

I think that authors know the genes after whole genome sequencing and not after conjugation experiments. Indeed, according to lines 206-207, only one of the 64 isolates, named LNp148, was used for conjugation experiments so is the only confirmed by this methodology.

Please, put the results data according the suggested arrangement.

 3) Preliminary comment: “Comments 12: Lines 202-205. Please, check the figures: 60 isolates tetAv1 + 6 isolates tetA-1 = 66 isolates. In addition, if the other four tetA variants where detected at least in one isolate, you have to have to add 4 + 66 = 70, but only 64 isolates were studied. So, the whole distribution of tetA variants among the 64 isolates must be added. How many isolates had the five tetA varians detected? Add data stratification by animal species.

Response 12: 6 isolates containing tet(A) variant gene also harboured tet(A)-1 gene, so the total number of isolates is still 64. We have added a supplementary table S4 with this information in supplementary materials.”

Many thanks for adding supplementary table S4. Nevertheless, it is a pity the text remains confusing (Lines 215-218: “Among the 64 tet(A)-positive E. coli strains, we identified a total of five tet(A) variants (designated tet(A)v1 to tet(A)v5), with tet(A)v1 to be most predominant, present in 60 of the 64 isolates. Additionally, tet(A)-1, the previously reported allele of tet(A), was also 217 found in 6 of the 64 isolates (Supplementary Table S4)”.

 I suggest a simpler paragraph: “Among the 64 tet(A)-positive E. coli strains tet(A)v1 was the most predominant, present in 60 of the 64 isolates. The remaining four isolates harboured four additional variants designated tet(A)v2 to tet(A)v5). Additionally, tet(A)-1, the previously reported allele of tet(A), was also found in 6 of the 60 isolates harbouring tet(A)v1.” Or a similar paragraph.

 4) Preliminary comment: “Line 121. Please, specify the meaning of “selected strains”.

Response 5: “selected strains” suggests that we randomly selected 30 representative strains to sequence, which has been revised in line 137.”

“Randomly” is probably not the best criterion for selecting isolates and covering the five mutations, having in mind that four of the five mutations were rare (only one isolate per mutation). Accordingly, add the data of the mutations covered to the paragraph in lines 269-283. In addition, lines 269-271 are confusing since seems that only these 30 isolates were sequenced, where it is clear from M&M section that 64 isolates were sequenced by Illumina technology and 30, additionally, also by MinION.

 5) Preliminary comment: “Comments 24: Lines 261-266. It is not clear how this information was obtained. Please, add the required data to M&M section. Where is the description of the new three tetA variants?

Response 24: We have a clear statement on lines 287-288.”

Lines 287-288 state “we identified eight tet(A) variants (including the previous five) in 103 E. coli isolates (4.2%, 95% confidence interval: 3.4-5.0%) originating from 13 of the 17 test provinces or municipalities.” But this is not the answer to my question. If at point 3.4 the authors present results about distribution of tetA variants, the M& section mentioned 64 isolates and the Results section five tetA variants, the readers are confused by the text “we identified eight tet(A) variants (including the previous five) in 103 E. coli isolates”. If preliminary data (published or unpublished) are used it can be clearly stated and added to M¬M section.

 6 ) Preliminary comment: “Comments 19: Lines 231. Having in mind the low number of isolates it is expected to found no statistically differences. Indeed, visual analysis of histogram “A” of supplementary figure S1 strongly suggest clear differences between the first column in comparison to the other two.

Response 19: There may be situations like the one you mentioned, but there really isn't a statistical difference”.

My comment was in the line that these results of supplementary figure S1 suggest that, if instead 5 versus 8 isolates, a higher number of isolates could be included into the study of relative expression of pumps, probably the authors could be able to detect differences among the groups in the case of figure D (ompF). This should be taken into account when the results are presented.

 New comments:

 Line 20. The sentence “the V55M mutation was unequivocally validated for its positive role in conferring tigecycline resistance” was no supported by date presented at lines 221-224, and especially the figures of table 1, since all, and not only V55M, has an effect over de TGC CMI. Differences of one step, for 1 mg/L to 2 mg/L, are not substantial. Curiously, the authors are more cautiousness at line 356: “which maybe relate to the non-synonymous mutation V55M

 Table 1. Some of the figures of the table (30, 50, 110, 7) are unusual. If “Minimum inhibitory concentrations (MIC) were determined using broth microdilution” (lines 93-94), the expected figures for two-fold dilutions are 0.25 – 0.5 – 1 – 2 - 4 - 8 -16- 32 – 64 – 128, etc. Could you explain, please?

 Line 248: Having in mind that 60 isolates had tet(A) v1, why only 23 isolates were used for relative expression assays?

 Line 282: Please, add the number of isolates instead of “certain”.

 Lines 319-321. The authors state that “the variability in tigecycline resistance observed

across these strains does not exhibit a strong correlation with the expression levels of th

tet(A) gene and its regulatory counterpart, tet(R)”. This is an interesting approach but no correlation analysis has been provided. Add the results of the correlation analysis to the results section before their discussion.

 Supplementary figure S3:

Please, check the axes because it seems there have some confusion.

A: tetA without drug

B: tetA without drug?

C: tetR with tigecycline

D: tetR with tigecycline?

 General comment about spelling. If the isolates analysed are not deposited in an international collection, it is better to use “isolate” instead of “strain” along the manuscript.
